# Cancer Stem Cells (CSCs), Circulating Tumor Cells (CTCs) and Their Interplay with Cancer Associated Fibroblasts (CAFs): A New World of Targets and Treatments

**DOI:** 10.3390/cancers14102408

**Published:** 2022-05-13

**Authors:** Beatrice Aramini, Valentina Masciale, Chiara Arienti, Massimo Dominici, Franco Stella, Giovanni Martinelli, Francesco Fabbri

**Affiliations:** 1Division of Thoracic Surgery, Department of Experimental, Diagnostic and Specialty Medicine—DIMES of the Alma Mater Studiorum, University of Bologna, G.B. Morgagni—L. Pierantoni Hospital, 47121 Forlì, Italy; franco.stella@unibo.it; 2Division of Oncology, Department of Medical and Surgical Sciences, University of Modena and Reggio Emilia, 41122 Modena, Italy; valentina.masciale@unimore.it (V.M.); massimo.dominici@unimore.it (M.D.); 3Biosciences Laboratory, IRCCS Istituto Romagnolo per lo Studio dei Tumori (IRST) “Dino Amadori”, 47014 Meldola, Italy; chiara.arienti@irst.emr.it (C.A.); giovanni.martinelli@irst.emr.it (G.M.); francesco.fabbri@irst.emr.it (F.F.)

**Keywords:** cancer stem cells (CSCs), circulating tumor cells (CTCs), cancer associated fibroblasts (CAFs), cancer

## Abstract

**Simple Summary:**

The world of small molecules in solid tumors as cancer stem cells (CSCs), circulating tumor cells (CTCs) and cancer-associated fibroblasts (CAFs) continues to be under-debated, but not of minor interest in recent decades. One of the main problems in regard to cancer is the development of tumor recurrence, even in the early stages, in addition to drug resistance and, consequently, ineffective or an incomplete response against the tumor. The findings behind this resistance are probably justified by the presence of small molecules such as CSCs, CTCs and CAFs connected with the tumor microenvironment, which may influence the aggressiveness and the metastatic process. The mechanisms, connections, and molecular pathways behind them are still unknown. Our review would like to represent an important step forward to highlight the roles of these molecules and the possible connections among them.

**Abstract:**

The importance of defining new molecules to fight cancer is of significant interest to the scientific community. In particular, it has been shown that cancer stem cells (CSCs) are a small subpopulation of cells within tumors with capabilities of self-renewal, differentiation, and tumorigenicity; on the other side, circulating tumor cells (CTCs) seem to split away from the primary tumor and appear in the circulatory system as singular units or clusters. It is becoming more and more important to discover new biomarkers related to these populations of cells in combination to define the network among them and the tumor microenvironment. In particular, cancer-associated fibroblasts (CAFs) are a key component of the tumor microenvironment with different functions, including matrix deposition and remodeling, extensive reciprocal signaling interactions with cancer cells and crosstalk with immunity. The settings of new markers and the definition of the molecular connections may present new avenues, not only for fighting cancer but also for the definition of more tailored therapies.

## 1. Introduction

In recent decades, cancer stem cells (CSCs) and circulating tumor cells (CTCs) have become a point of attraction, both in basic research and clinical studies, due to their multiple roles and involvement in tumorigenesis, cancer progression and therapy resistance [1,2]. However, it is currently difficult to exploit these cells in a clinical setting, firstly due to a lack in fully-validated biomarkers, and second due to a non-unique method for their identification. In fact, there is still not a specific single marker or a combination of markers which represents these subpopulations of cells clearly distinguishing them from the conventional cancer cells. Indeed, it is known that CSCs constitute only a small percentage (0.05–1%) [3,4] of tumor cells. Additionally, CSCs and normal stem cells also share similar transcription factors and signaling pathways. Consequently, it continues to be more challenging to isolate and identify CSCs. In a similar way, despite being functional for noninvasive-liquid biopsy, CTCs cannot be considered a full diagnostic tool due to the controversy over their clinical utility. In addition to these unsolved aspects, the roles of the tumor microenvironment are still largely debated due to the difficulty of identifying the main components and the molecular pathways connected with cancer development and dissemination. In this context, cancer-associated fibroblasts (CAFs), one of the key-components of the tumor microenvironment for their multiple functions (e.g., matrix remodeling and capability to modulate the crosstalk between cancer cells and tumor immunity), may be considered additional possible co-actors, and then targets, in combination with CSCs and CTCs [5,6]. Although with difficulties, the identification of new markers and pathways connecting and/or characterizing these populations is a future challenge for the development of tailored therapies against cancer.

## 2. Cancer Stem Cells (CSCs) and Their Roles in Tumor Formation and Dissemination

The scientific community has recently discovered several processes for new treatments of cancer [7,8]; however, the disease still remains an issue for the scientific community all over the world [9]. In particular, focus has moved to the “early stages” of cancer and may have a high chance of finding a resolutive cure through surgery or surgery plus medical treatments. In addition to the fact that, for most patients, the diagnosis comes at an advanced stage where dissemination to other organs is already present, in the early stages, there is a risk of recurrence that seems to be related to small elements such as cancer stem cells (CSCs) [10,11]. These have been proposed to be the causes of cancer progression, metastasis and drug resistance [12]. Recently the scientific community has discussed specific aspects of these cancer related actors, considering them as one of the most interesting and unknown causes of tumor dissemination [13,14]. In particular, they seem to be present in both early and locally advanced stages, as shown by Masciale et al. for adenocarcinoma and squamous cell carcinoma of the lungs [15]. In that research, a pool of overexpressed genes was related to recurrence [16] at different NSCLC stages. Those genes need to be investigated further as they could open up new frontiers for targeted therapies. It is critical that these research paths are taken, as CSCs may resist the common medical treatments, leading to tumor aggression [17,18].

In recent decades, theories related to CSCs have generated greater interest [19,20]. Scientists hypothesize that these cells will revolutionize our understanding of the cellular and molecular events related to cancer development and dissemination [12,14,21]. Considering tumor progression theory, among all the tumor cell populations, these cells are placed at the top of the hierarchy for their specific characteristics and high proliferative potential [22,23]. CSCs are close to normal stem cells in terms of their cellular properties, such as their capacity to self-renew and rebuild the progeny [24,25]. One of the most widely discussed aspects of these subpopulations of cells is our capacity to identify specific markers, which has not yet been completely achieved by the scientific community due to difficulty with isolating markers for certain cancer cell populations [23]. So far, the markers commonly used to identify them are not specific to these subpopulations of cells but are markers associated with stem cells in general, such as superficial antigens such as CD133, CD44 and CD90, or intracellular enzymes such as aldehyde dehydrogenase (ALDH) [26,27,28].

It is also clear that one marker alone is not enough to identify this exclusive population. It is necessary to consider multiple markers in combination [11,29]. As an example, in 2018, Satar et al. [30] sorted two subpopulations of NSCLC known as the triple positive (EpCAM+/CD166+/CD44+) and triple negative (EpCAM-/CD166-/CD44-) subpopulations by fluorescence-activated cell sorting (FACS). In their study, they showed that the triple positive subpopulation demonstrated similar characteristics to CSCs when compared to the triple negative subpopulation. In 2019, Masciale et al. searched for the ALDHhigh cell population using CD44+/EpCAM+ marker cells, demonstrating that these markers can identify two subpopulations of ALDHhigh cancer stem cells that are highly correlated, although the same has not been found when using only the superficial marker CD44 and ALDH [29]. Nonetheless, the tumorigenicity of these cells was previously demonstrated in immunodeficient mice, in which CSCs were transplanted with the result being growth of a tumor formation [12]. Moreover, these cells have been tested in vitro for solid tumors such as lung, brain, breast, colon, etc., with the formation of neurospheres, mammospheres and colonospheres, respectively [28,31,32]. Although the origin of CSCs and their phenotypes and functions are still debated [14,33], based on their proliferative capacity and lack of sensitivity to chemotherapy, CSCs may represent quiescent, non-dividing cells with the potential to activate resistance mechanisms [14,34].

Though still not fully proven, the tumor microenvironment plays an important role in tumor dissemination and metastatic processes. It is presumed to support the cellular processes of CSCs [12,35]. Indeed, in this context, several actors (molecules, vesicles, normal cells) seem to interact, directly or indirectly, with CSCs, supporting CSC maintenance/perpetuation, inducing chemoresistance, and eventually promoting angiogenesis and tumor cell escape from the lesion [36,37,38].

## 3. Circulating Tumor Cells (CTCs): Drivers of the Metastatic Process

CTCs are the cancer seeds that disseminate from the primary tumor and/or from metastatic lesions, causing tumor progression and potentially endlessly propagating the disease. They mainly circulate in the peripheral blood, lymphatic vessels, and into the stroma of the starting or metastatic lesions, representing the liquid stage of a solid tumor (Figure 1). Indeed, CTCs are vital components of liquid biopsies and have been under investigation for the identification of residual cancer, monitoring of therapy response, and prognosis of recurrence and survival [39,40,41]. Constituted by a pool of rare and highly heterogeneous cells, able to change their antigen expression, even during their travel through the body (e.g., thanks to Epithelial Mesenchymal transition, EMT) [42], they can mirror the disease characteristics and its evolution. Hence, researchers studying their molecular features are willing to prove their clinical utility for identifying treatment resistance mechanisms in patients. Even in the development of personalized medicine for non-small cell lung cancer; in fact, CTCs could play a crucial part in the diagnosis, prognostication and choice of treatment. Indeed, on these bases, Luo et al. [43] expressed the now acquired concept that conventional treatment strategies might not be fully useful in patients with metastasis: standard treatment approaches are based on the pathological and molecular characterization of the primary tumor, potentially sparing metastatic evolution and selected cells and tissues. Hence, CTCs analysis in patients with lung cancer could serve as a useful surrogate marker of distant metastasis and offer the possibility of monitoring changes in tumor genotypes and phenotypes during treatment, helping clinical management. Despite they have to be still validated before routine clinical application, CTCs have been detected and studied since years before and after surgery in patients with NSCLC, with a particular interest in tumor dissemination and prognosis, disease and therapy-response monitoring, they have been considered a novel factor that could help predict the risk or recurrence [44,45,46]. CTCs count was different but progressively increasing in numbers through disease advanced stages and always correlating with a poor prognosis and scarce curative therapeutic effects [45]. Furthermore, CTCs molecular characteristics have strengthened previous findings. Huang M et al. [47] reported aneuploid CTCs correlated with both radiological and pathological responses in patients with NSCLC who received neo-adjuvant approach, suggesting that enumerating and karyotyping can serve as a surrogate marker for disease monitoring in that stage of NSCLC. Fina E et al., [48] showed that CTCs occur frequently in NSCLC patients, even without evidence of distant lesions; hence, CTCs sub-populations could be used as prognostic biomarkers. Marianna Gallo et al. [49] summarized that in early and advanced NSCLC, the detection of CTCs has potential for early diagnosis and prognosis. CTCs can express PDL-1, CD44 of carry EGFR and Kras mutation. In particular, CD44 has been related to chemo sensitivity in primary CTCs cultures and in certain isoforms it has also been related to CSCs, suggesting an additional connection between CTCs and CSCs. PDL-1 has been associated with a worse/shorter OS (overall survival) and PFS (patient free survival). Furthermore, the nowadays feasible genetic characterization of CTCs has demonstrated that it is feasible to detect KRAS and EGFR mutation status in lung cancer utilizing these cells as liquid biopsy. Summing up, EMT, PDL-1 and mutation, or other molecular features, present or not in the primary tumor, and probably even in the CSCs, can also be detected in CTCs. Since these characteristics are almost always related to poorer prognosis and scarce therapeutic effects, the importance of CTCs and their link to CSCs is clear. Lastly, it has to be noted that oligo-cluster of CTCs are associated with even worse unfavorable clinical outcomes and a 20- to 100-fold greater metastatic potential than single CTCs, involving that, in addition to a molecular correlation, even a physical connection between tumor and nor-mal cells has a role in tumor spreading. All of these results/research indicate CTCs molecular analysis and the study of their correlation with CTCs and normal cells could represent a ground-breaking approach. It could help clinicians where the utility of solid biopsy is strongly limited because of tumor heterogeneity and repeated tissue sampling is not possible. Moreover, it can reveal dynamically and non-invasively the complete molecular landscape of a tumor and of its involvement with the normal and tumor microenvironment [50]. Hence, it is becoming more and more clear that a single biomarker or assay will always fail to reveal the full nature the neoplasia, being unable to depict the complexity of a tumor alone, and its growth and progression. A multiparametric/multi-omic approach on different starting material has to be favored. In particular, it can be hypothesized that CTCs research should be undertaken simultaneously with CSCs and other microenvironmental cells and factors, such as CAFs, cytokines, and extra-cellular vesicles. The multi-omic analyses of these elements is absolutely worth being performed and should become a standard approach.

## 4. CSCs and CTCs: Connections and Interactions with Cancer

The connections and interactions between CSCs and CTCs are still not completely clarified; indeed, it is mandatory to define the peculiarity and characteristics for each subpopulation of cancer cells and the links among them [43,51,52]. The main question is set on the possibility to predict recurrence in solid tumors [53,54]. In fact, in addition to the multimodal approaches commonly used by the oncologist, which may include surgery, chemo- and/or radiotherapy in early or locally advanced patients, the cancer relapse still represents an unsolved problem and the principal cause of cancer-related death [54,55]. The risk of recurrence has led scientists to identify a prognostic score in solid tumors, although so far, they are yet to find a score that is able to predict cancer relapse [16,56,57]. There are many causes of tumor recurrence; systematic reviews have highlighted a possible link between cancer cells and dynamic clonal evolution in common medical treatments, including radiotherapy [58,59]. Moreover, the resistant clones of tumor cells develop somatic mutations and variations not common with the origin state of these cells [60]. To overcome these problems, the surface markers used to identify and isolate cancer stem cells are objects of interest for the possibility to dissect inter- and intra- tumors’ heterogeneity more rapidly [11,28]. Notwithstanding this possibility, these heterogeneity types can be even better verified with a NGS-based multi-omic approach [61,62]. This way could reveal even more information regarding the potential CTC–CSC connection, uncovering pathways of resistance and metastatic processes [63].

Another potential link concerns the site of cancer relapse [55]. It seems to depend on the tumor microenvironment, which provides a rich adaptive landscape for tumor cells’ dissemination. It has been shown that the adhesion of cancer cells to the extracellular matrix (ECM) activates specific genes that promote cancer progression or induce a quiescent status [64,65]. Identifying these genes and their expression could be useful to prevent tumor recurrence, e.g., helping to unlock mechanisms of CTCs spreading, thus improving patients’ long-term survival [66]. Recently, the most overexpressed genes were described as related with cancer recurrence in NSCLC stem cells for early and locally advanced patients underwent surgery [15]. This study highlighted the roles that CSCs genes may play in future targeted therapies and deeply analyzed their potential connections in the tumor microenvironment. Moreover, a recent prospective cohort study showed a positive correlation between CSC frequency and the risk of relapse in locally advanced NSCLC patients [67]. These results highlight the need for further molecular investigations about the prognostic role of CSCs at different lung cancer stages [67].

Together with CSCs, CTCs are the other important cell subpopulation related to recurrence. Like CSCs, CTCs can show multiple phenotypes, a dynamic range of antigen expression and genetic, genomic and transcriptomic alterations, most of them related to EMT [68,69,70,71,72], that can change in time and space depending on the overall tumor nature, the nearby microenvironment and the patient’s condition. Although the common origins or their relationships of CTCs and CSCs have not yet been demonstrated, some hints of scientific evidence follow this possibility. Markers such as EMT-related and CD133, CD44 EpCAM, and ABC-G2 can be shared between these two populations, as well as DNA and RNA alterations [73,74].

Patients with chemo- or radio-resistance have been shown to have high counts of EMT-transformed CTCs [67,68], which seems to influence cancer patients’ survival [75,76,77]. It has been also suggested that an increased number of CTCs and their downstream transformation into CSC-like cells may induce cancer relapse [52]. In particular, the chemo-or radio-resistance showed elevated numbers of EMT-transformed CTCs, whereas clinical studies correlated poor survival with cancer patients linked to EMT phenotypes in tumor cells [75,76,77]. Furthermore, identifying the biunivocal connection linking EMT with CTCs/CSCs is of great interest for understanding cancer progression and the potentiality of CTCs to build metastases fairly similar to the primary tumor, but at the same time more resistant and able to continue spreading.

## 5. Cancer-Associated Fibroblasts (CAFs): The State of the Art

Cancer-associated fibroblasts (CAFs) are a key component of the tumor microenvironment with fundamental functions based on extracellular matrix formation, stroma and metabolism modifications, cancer vessel neoformations and crosstalk interactions between tumors and immunity [78]. Understanding the diversities and connections between normal and tumor fibroblasts may be helpful to better highlight the roles CAFs play in cancer hallmarks, with particular attention to CAF subpopulations and their functions in cancer progression [79,80]. In 1889, Dr. Paget developed a theory based on the possible progression of the metastatic process, defined as “seed and soil”. Paget defined this “soil” as an active process related to tumor characteristics and set on progression and metastasis [81,82]. The “seed” has been studied repeatedly in the past few decades and now it can be represented by residing cancer cells, CSCs and even by CTCs [40,83,84], whereas the “soil” is composed by the tumor microenvironment (TME), and non-tumor actors, i.e., fibroblasts, immune cells, endothelial cells, proteins and tumor-inducing factors such as cytokines, growth factors, extracellular vesicles, etc. [85].

Discerning the interactions between cancer and the TME is considered necessary to better understand the roles of CAFs in the TME as causes of tumor progression and ECM remodeling. In particular, e.g., the depletion of CAFs induces pancreatic tumors to behave aggressively [86,87]. It is well known that CAFs are a heterogeneous population of cells with pro-tumorigenic and anti-tumorigenic functions [88]; investigating these characteristics may lead to new perspectives in terms of future targeted therapies. Furthermore, this could potentially clarify some of the roles of EMT as it has been suggested as a cancer cell mechanism to drive functions as tumor development and progression, together with CAFs in certain conditions, including a strict crosstalk between cell populations [89]. Furthermore, the grey area between inflammation and tumor is of great interest. In non-malignant conditions such as injury or inflammation, the resident fibroblasts that are activated are normal activated fibroblasts (NAFs) or fibrosis-associated fibroblasts (FAFs) [78,90]. Yet, the answer to the insult (i.e., the activation of NAFs) ends as soon as the insult does. The mechanisms of NAFs’ activation are similar to those of CAFs in terms of tissue remodeling and immune stimulation, although the mechanisms behind the transition from NAFs to CAFs have not been well-defined by the scientific community [90]. Looking ahead, identifying the differences or common characteristics between fibroblasts in cancer or inflammation will be helpful for understanding the different roles of CAFs and NAFs in cancer [90].

Several studies have shown that tumor-derived CAFs are able to drive the EMT phenotype in cancer cells [91,92,93]. In particular, in colorectal cancer cells CAFs derived chemokine CCL2 activate the fibroblast receptor 4, which induces b-catenin promoting EMT [92]. In prostate cancer, CAFs secreting CXCL12, through the conversion of EMT phenotype, increase the expression of CXCR4 which promotes the cancer migration and metastasis in vivo [93]. This phenomenon has also been described in breast cancer for the contribution of CAF to induce metastasis through the enhancement of stem cell features in cancer cells [94] by the activation of the Notch mechanism, with the consequent upregulation of CSCs markers as, for example, the aldehyde dehydrogenase (ALDH) 1A3 [94]. Interestingly, the development of fibrosis induced by myofibroblasts and myofibroblast/collagen I has shown to be protective against the development of cancer in the pancreatic gland [94]. This may suggest the pro-tumor role of CAFs against cancer. In particular, myofibroblasts and the fibrosis depletion in mouse models are also associated with the acquisition of EMT features, increased hypoxia and a high number of CD44+/CD133+ CSCs, as well as impaired immune response. Lastly, mice showed more invasive tumors, features associated with specific and unfavorable genetic aberrations [94].

## 6. Extracellular Vesicles (EVs)

Extracellular vesicles (EVs) are particles released from the cell that are enclosed by a lipid bilayer and cannot replicate [95]. In general, since no consensus has emerged on specific markers of EV subtypes, it is not entirely possible to use a specific classification. Operational terms for EV subtypes, such as small- and large-EVs, should be preferred [95]. EVs are key mediators of communication between cells in normal physiology and pathology. They orchestrate many systemic pathophysiological processes, such as vascular leakiness, coagulation, and reprogramming of stromal cells to help pre-metastatic niche formation and subsequent metastasis. Hence, the prognostic and functional importance of tumor-derived EVs (tdEVs) is clear and has already been reported [96]. A growing number of results and literature indicate that they could be utilized for early cancer detection, prognosis, therapy efficacy prediction, and even to plan innovative therapeutic procedures. Indeed, their role in tumor cells and stromal cells interplay, in local and distant microenvironments, plays a critical part of both primary tumor growth and metastatic evolution. Evidence reported in recent decades involves tdEVs in neo-angiogenesis, enhancement of proliferation, migration, and growth of endothelial cells, micro-emboli formation promotion, ECM remodeling, EMT triggering and induction, priming of normal tissue for cancer colonization, involvement in organotropism and niche formation events, and education of cancer cells toward therapeutic resistance [97]. Thus, clinically, tdEVs may be biomarkers and novel therapeutic targets for cancer progression at the same time, particularly for predicting and preventing future metastatic development [98]. EV roles are carried out thanks to their cargo. Indeed, they are selectively supplemented with a number of cellular bioactive molecules: DNA, ncRNAs, e.g., miRNAs, and surface proteins (integrins, RNA-binding proteins, signaling receptors), sometimes characteristic of the cell of origin, involved in the recognition of and EV-uptake by receiver cells, in the transfer of therapy resistance, and in the generation of an immunosuppressive microenvironment [99]. Hence, the likely involvement of EVs in many aspects of the metastatic processes, potentially being involved in CSC intrinsic capabilities, CTC acquired skills, and in the communication between CSCs, CTCs, and CAFs is clear [37]. This is why EVs have emerged as prospective biomarkers to monitor cancer growth and progression and as potential therapeutic targets [100].

Examples of these assumptions are many. PDL-1 on EVs in plasma are elevated in melanoma and HNSCC and correlate with non-responders to anti-PD1 treatment [101,102]. In pancreatic cancer, EV surface presents B cell antigens that induces a reduced cytotoxicity against cancer cells, thus strengthening the immune-suppressive role of EVs in cancer [103,104]. EVs from lung adenocarcinoma cells can transport miRNA correlated to higher invasive potential, explicated through the promotion of the Wnt/catenin pathway and EMT, and inducing cell proliferation [105]. Lung cancer cell EVs can trigger fibroblasts expressing pro-angiogenic factors such as IL-8 and VEGF, which in turn stimulate both lung cancer growth and attract endothelial cells [106]. EVs from SUR1-expressing lung cancer cells can induce CAFs and promote fibroblast migration and promote cisplatin resistance [107,108]. Lung CSC-derived EV miR-210-3p contributes to a pro-metastatic phenotype [109]. Hence, EVs are showing progressively to be involved in cancer progression and resistance, metastasis, and even in the intercommunication between CSCs, CTCs and CAFs.

## 7. CAFs, CSCs and CTCs: Interconnections and Mechanisms of Metastatic Tumor Dormancy

The metastatic process is one of the most debated and presently unsolved factors affecting every solid tumor. In particular, the tumor microenvironment (TME) and the interactions between cancer cells and non-cancer cells are crucial for cancer progression [110,111]. Actors such as CAFs, CSCs and CTCs (Figure 2) can be considered as a main point of connection between primary cancer and recurrence that remains to be clarified [2]. Looking at each of those, it has been suggested that a population of circulating CSCs (cCSCs) could be considered to be “more metastatic”, more dangerous, than “conventional” CTCs, and CAFs seem to be the mediators of both subpopulations. Indeed, CAFs exist in the blood both as individual cells and as complexes with CTCs. It is known that the metastatic potential of CTCs, and, hence, of cCSCs can be improved by clusters of CAFs known to promote cancer invasion and dissemination [112,113]. These complexes help CTC survival in the blood and probably help them in colonizing the pre-metastatic niche. cCAF/CTC clusters were recently identified and counted in the blood of cancer patients and then associated with clinical features and outcomes [112]. Furthermore, as a further confirmation, if CAFs were injected, along with MCF-7 cells, into NSG mice, an increasing number of cCAF/CTC clusters is correlated with tumor grow and metastasis [112].

Similar results have been shown in breast cancer cells where enriching stem cells from MCF7 mammospheres resulted in CAF/CSC clusters in vitro. In mice co-injected with non-metastatic MCF7 cells and CAFs from a TNBC/Basal-like BC (CAF23), metastatic disease was shown and enrichment of cancer stem cell (CSC)-like CTCs, triggering the presence of circulating cCAF/MCF7-CSC clusters. Destroying cCAF/CTC complexes, as well as clusters with other type of cells [114,115] may open up new frontiers for controlling cancer or preventing metastasis.

In particular, an interesting concept is that of cancer “dormancy”, a clinical condition in which tumor cells are occult, asymptomatic and indiscernible for a certain period, which is probably a further trick by which the tumor develops a resistance to medical treatments [116,117]. Dormancy has been studied in several solid tumors such as breast, liver, prostate, etc. [118,119,120,121,122,123]. However, CSCs, CTCs and CAFs can be related to this phenomenon as subpopulation of cells able to induct recurrence, enhancing the metastatic process. In fact, one of the most problematic aspects of cancer is its ability to recur after apparently successful primary treatment. In some cases, these recurrences can happen years or even decades after initial diagnosis and treatment. However, evolving technologies for monitoring the dormancy and micrometastatic diseases, including CTCs and other blood biomarkers, may promote the ability to detect small volumes of residual cancer. This concept is still quite far from real clinical setting application for CSCs and CAFs which need to be deeply investigated. However, this theory of dormancy contradicts the notion to define tumor genetic signatures for the difficulties to dormant cells to be responsive to various stimuli exploiting cell cycle, kinase signaling and epigenetic features in single cell data. This may suggest the necessity to discover a “signature of dormancy” to explain the therapeutic resistance.

Recent translational studies and genomic sequencing have shown several links to dormancy in solid tumors [124,125,126,127]. In particular, in breast and prostatic cancer, signature genes such as NR2F1, SHARP1, BMP7high and COCOlow seem to be related with dormancy and reactivation in specific metastatic sites [128]. Recently, EMT-transformed cells have been related to a low induction of proliferation [129,130,131], which seems to be regulated by an EMT program that drives cancer cells to dormant CTCs. For these reasons, it is important to identify the molecular processes occurring between EMT-positive CTCs and CSCs, especially during the quiescent stage, to define whether these cells start the relapse process or remain stable in a quiescent form. In this context, the tumor microenvironment in certain organs such as the lungs, liver, brain, etc., plays a role as “promoter” of dormancy [129] by preserving reciprocal signals between CTCs and CSCs and by activating the expression of pro-dormancy genes. These extracellular signals can be cytokines and/or EVs. Furthermore, it has to be noted that these two cellular sub-populations can coexist in a niche with protection from the immune system, which can extend the dormancy period.

Today, there are examples where drug-induced dormancy signals protect the body from recurrence, such as, for example, in multiple myeloma, when bortezomib induces cancer cells apoptosis [to become quiescent] [130]. Moreover, a potential contributor to therapy resistance in this case derived from the inhibition of eIF2α-dephosphorylation when bortezomib induces by the GADD34-PP1c inhibitor which helps bortezomib to eliminate dormant cancer cells thus controlling relapse [130]. The potential for a similar scenario has been investigated in other organs, e.g., breast cancer, where low expression of extracellular signal-regulated kinase (i.e., ERK) and high levels of p38α have been shown in dormant tumor cells [131]. In this regard, the protein p38α activates p53 (R213Q), BHLHB3 and NR2F1 and inhibits FOXM1 and c-JUN, connected with the G1-S transition [131,132]. Genes’ dormancy signatures have been studied and found to delay recurrence in breast and prostate tumors [133], although inhibition of p38a in vivo has been shown to possibly preserve tumorigenicity, even in dormant cells [134]. In further research on head and neck cancer, scientists have demonstrated that transforming growth factor-β 2 (TGFβ2) is induced by dormant cells [134]. TGFβ2, through TGFβRIII, induces SMAD1/2/5 to upregulate p27 with the consequent activation of p38 for dormancy [134]. Moreover, the structures of the ECM determine cells’ proliferation or dormancy through the mediation of intracellular pathways, which may stimulate cancer cells’ progression, dissemination or metastatic processes’ development. In particular, downregulation of the urokinase receptor in squamous cell carcinoma (HEp3) keeps α5β1-integrins inactivated [135] with the consequent inhibition of focal adhesion kinase (FAK), which leads cells to bind to fibronectin.

The opposite process, which induces dormant cancer cells’ activation in a tumor microenvironment, activates a cascade of signaling such as collagen I-mediated integrin β1 signaling and that of Src and FAK for the phosphorylation of myosin light chain kinase via the ERK pathway [136]. This process is important as cytoskeletal rearrangements in the ECM composition are crucial for determining dormancy or inducing tumor metastasization. The future inhibition of growth-inducing structural modifications of the ECM-associated microenvironment may be key to highlighting new approaches to preventing recurrence.

## 8. Tumor Microenvironment and Immune System in NSCLC

Lung cancer is a leading cause of death around the world, due to its poor prognosis and 5-year survival rate [137]. Target therapies and immunotherapy in particular, which have been developed in the last few years and are considered promising treatments in a selected population, have improved the survival rate and quality of life in cancer patients [138,139]. However, these treatments are not completely effective, and several studies have demonstrated the lack of or incomplete immune response in solid tumors, for an overall response rate of 15–20% [140,141]. The scientific community is moving toward defining other targets or molecules that may improve the response against cancer, with particular interest in finding a better definition of the unknown mechanisms between the immune system and cancer [142].

Recently, the role of tumor microenvironment (TME) in non-small cell lung cancer (NSCLC) has been extensively studied, and has been found to contribute to tumor development, homeostasis maintenance, and cancer progression [143,144]. In 2016, Carbone et al. found a large number of Treg cells in a tumor, which are specialized to suppress immune response and maintain homeostasis and self-tolerance [145]. It has been shown that Tregs are able to inhibit T cell proliferation and cytokine production and play a pivotal role in the maintenance of peripheral tolerance and, therefore, in preventing excessive immune responses and autoimmunity [146,147]; moreover, the high number of CD8+ T cells increased PD-1 expression, which was an expression of impaired immune function [147,148].

Furthermore, several results from clinical studies in NSCLC patients highlighted the connections between the immune system and the clinical outcomes [149,150]. In particular, the identification of “preformed” antitumor T cells and antibodies in NSCLC patients’ blood are of great interest, in addition to the presence of tumor-infiltrating lymphocytes (TILs), mainly characterized by CD8+ T cells, which have been highly correlated with a better survival rate and tumor staging characteristics [151,152]. The presence of TILs at early stages seems to be a favorable factor on prognosis and recurrence [153]. Moreover, the presence of both CD8+ and CD4+, and in general CD3+ T cells associated with a high presence of mature dendritic cells (DCs) has been associated with improved survival [154,155]. Contrarily, a high number of Treg cells and M2 macrophages seems to predict decreased survival, similar to the presence of high PD-L1 expression [145,148,156]. The association between cancer and inflammation is defined by scientists as “a circle process”, due to the fact that an inflammatory response may activate stromal and cancer cells with a direct infiltration of immune cells at different immunosuppressive and protumorigenic stages [157,158]. The polarization mechanism can involve not only tumor-associated macrophages (TAMs) [159,160], but also tumor-associated neutrophils (TANs) [160], myeloid-derived suppressor cells (MDSCs) [161], B cells [162], and T cells [163], which may be able to turn down the adaptive immunity and have an impact on cell fate and the patient’s prognosis.

In addition, recent studies have also shown the capacity of immune cells to activate downstream stem cell pathways, with a consequent enhancement of self-renewal and development of resistance to common medical treatments [18,157]. It has been shown that, in a pancreatic tumor, TAMs acted as supporters of CSCs through the activation of STAT3, which is involved in cell survival [164,165]. The depletion of M2 macrophages, on the other hand, induces a reduction of CSCs with a consequent improved response to chemotherapy, although the combination with TANs and TAMs seem to implement the chemosensitivity with better outcomes [166]. In addition, cancer is able to evade the adaptive immunity through the upregulation of programmed death ligand 1 (PD-L1), and for this reason the immunotherapy is based on immune checkpoint inhibitor to act against that mechanism to inhibit the molecules that may downregulate or block the immune response [167]. To this positive role of immunotherapy should be added the highly tumorigenic effects of CSCs that negatively modulate the system, as they seem to interact with immunotherapy by upregulating the adaptive immune checkpoint PD-L1 [167,168], thus being a major cause of resistance, driving T-cell transfer, depletion and relapse. These findings showed that some cells, which are part of the immunity or of the tumor, may have a double role as both the cause of resistance to common treatments, including immunotherapy, and as future targets against solid tumors [167,168,169]. The connections between CSCs and immune cells need to be further investigated in terms of their role in TME and induction of the epithelial-to-mesenchymal transition (EMT). The use of TME modulation to control CSCs with the consequent blocking of tumor development and dissemination may be the key to improving responses to common therapies [170,171]. Moreover, EMT also interacts with CTCs, which are considered the indicators of residual cancer, or relapse [35,36]. Although CTCs are known as strategic blood micro-molecules that are involved in the metastatic cascade with a high prognostic impact on patients’ survival, the mechanisms of the escaping process of CTCs to the immune system are largely unknown [172]. Mesenchymal CTCs are usually a single cell or cluster, expressing EMT regulators, with the ability to regulate the response to therapies and disease progression. In particular, the presence of a hypoxic tumor microenvironment and EMT seem to increase the tumor variability and dissemination [173,174,175]. These unknown aspects highlight the urgent need to understand the processes among CSCs, CTCs, EMT, TME, and immune cells, which may be a milestone to further discoveries that improve the survival rate and quality of life in oncological patients.

## 9. The New World of Anti-Cancer Treatments

The history of cancer started a millennia ago with Egyptian and Greek populations, where patients with malignant neoformations were treated by radical surgery and cautery with very poor results and consequent death [176]. From this ancient Era to more modern centuries, several discoveries have been made, highlighting the biological and pathological characteristics of cancer for different organs with the consequent development of effective methods against solid tumors, such as the discovery at the end of the 1800s of X-rays. In fact, their utilization has become a pillar in medical oncological history, achieving the cure of several cancers [176].

After the Second World War, an improvement in cancer discoveries led to the identification of cytotoxic antitumor drugs and the birth of chemotherapy with an exponential development of research studies concerning the identification of new treatments. At the beginning of the 1980s, scientists started to think about cancer solutions as the discoveries of new markers targeting tumor growth and spreading processes [176]. This new approach opened up new “targeted therapies” which improved the patients’ survival, as the subsequent development of the genetic engineering process, which followed the advancement of clinical oncology and pharmacology. In particular, the definition of monoclonal antibodies and immune checkpoint inhibitors improved the patients’ survival and the quality of life, although complete remission has not been obtained in all the tumors. In fact, nowadays, the numbers of deaths for cancer are still high with a total number of 18 million new cases diagnosed in 2018 (2020), 19.3 million new cancer cases, and almost 10.0 million cancer deaths [177], of which the most frequent are lung (2.09 million cases), breast (2.09 million cases), and prostate (1.28 million cases) cancers. Concerning mortality, cancer is the second worldwide cause of death (8.97 million deaths) after ischemic heart disease [178]. Recently, the aim of defining molecules able to destroy cancer has encouraged researchers to study the development of new therapeutic approaches for cancer treatment, as cell therapies, anti-tumor vaccines, and new biotechnological drugs. These new perspectives have led to better patients survival from cancer, which has proceeded to increase across high-income countries, although international disparities persist [179]. The reduction in mortality for cancer reflects the persistent progress in oncological treatments against the so-called “big killers” as cancer of lung, breast, colon, prostate, etc. However, some types of tumors, such as lung and pancreatic cancers, still have high mortality rates. These data show that cancer has not yet been completely defeated and it represents a serious global health problem [180].

Beside the fact that surgical demolition is considered ineffective in patients with advanced stages, even for the early stages, which are the most dedicated to surgery, the solution is not definitive if we consider that for lung cancer the five-year survival rate is 56 percent for cases detected when the disease is still localized (within the lungs). However, only 16 percent of lung cancer cases are diagnosed at an early stage. For tumors that spread to other organs, the five-year survival rate is only 5 percent [181,182].

The epochal change was reached in the mid-1900s with the first revolutionary pharmacological approach represented by the development of the cytotoxic drugs against the tumor, although the cytotoxicity associated with the consequent drug resistance is still a serious obstacle to overcome [183]. Subsequently, more molecular approaches have been studied by the scientific community with the setting of the definition of the DNA structure and the development of new molecular techniques for DNA analysis. These approaches led to the discovery of specific gene alterations responsible for tumor development and progression [184].

Among these, monoclonal antibodies and new immunotherapeutic drugs have allowed the development of so-called “personalized medicine”, in which new therapeutic protocols are tailored to the patient, with higher efficacy and lower toxicity [185,186]. In addition, research in the field of oncology is constantly aimed at the discovery of new and effective therapeutic strategies, including the promising CAR-T Cell therapy, gene therapy (Yescarta and Kymriah) [187,188]. Simultaneously, research regarding new markers, targets, and where to find them, has increased in the last two decades. Indeed, the definition of new actors in cancer, as for example CSCs or in blood the CTCs and extracellular vesicles (EVs), lipid bound vesicles secreted by cells into the extracellular space and that can transport cargo—including DNA, RNA, and proteins—between cells as a form of intercellular communication [189], has opened new paths of research and clinical approaches. Beside the fact that these elements need to be better identified and studied for each tumor, scientists have started to give them more attention for their roles in tumor development and progression, associated with their possible roles in tumor resistance to therapies and consequent recurrence, which is one of the big obstacles for the efficacy of oncological treatments.

The definition of the roles and associated mechanisms of CSCs, CTCs, CAFs and EVs will certainly ease the identification of prognostic markers able to predict which patients are at a higher risk of recurrence after surgery, new elements predictive of resistance to therapies, and finally improve the survival of patients. In fact, the development of more tailored therapies associated with further clinical trials will increase the treatment efficacy and reduce the possibility of developing pharmacological resistance [190,191].

Particular attention should be paid to the development of new drugs. By now, in silico bioinformatics analysis directed toward the detection of the best molecule that can interact with a specific target can no longer be overlooked. Indeed, it is now possible to simulate the level of interaction of hundreds of new compounds with specific receptors, antigens and targets of the new drug. In this scenario, artificial intelligence and bioinformatics will be essential for the set-up of several in vitro and preclinical animal models to foresee the activity and the toxicity of the new drug and its therapeutic window. Therefore, today, bioinformatics and preclinical studies are the fundamental steps to develop a new effective drug endowed with the highest potential efficacy. The in silico and preclinical screening of thousands of different pharmacological molecules has, in fact, allowed researchers to obtain new oncological drugs, which are currently used in clinical practice, while significantly reducing mortality from oncological diseases [192,193].

## 10. Conclusions

Hence, it is mandatory to highlight again that, from surgery to systemic therapy, from histology to molecular characterization, from tumor tissue markers to liquid biopsy approaches, from in vitro preclinical experimentation to in silico drug development, and to a comprehensive omic analysis, it is now clear that a combined approach in translational oncology is absolutely mandatory and should become a standard approach. and may help researchers and clinicians to overcome this disease, encompassing most, if not all, of the cancer spectrum.

## Figures and Tables

**Figure 1 cancers-14-02408-f001:**
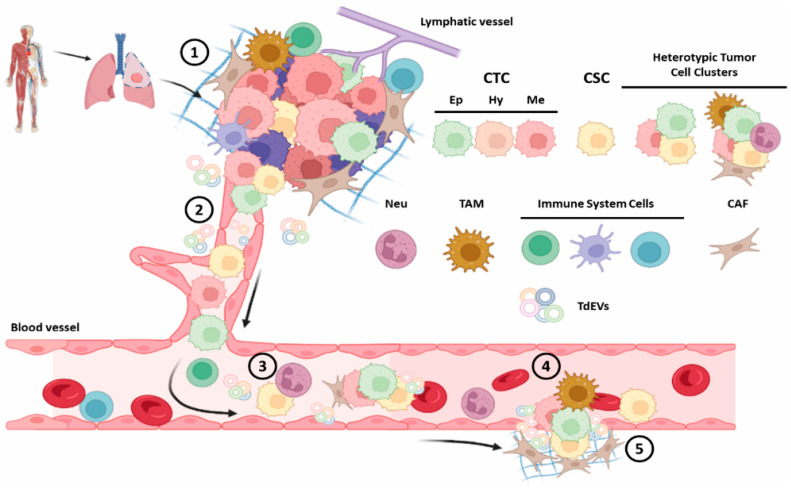
Representative image of principal metastatic steps and key actors. (1) Primary and/or metastatic tumor onset and growth, (2) intravasation and (3) survival into the blood flow, (4) extravasation at distant sites, and (5) expansion, dormancy or progression; circulating tumor cell (CTC), epithelial, mesenchymal and hybrid CTC (EpCTC, MeCTC, HyCTC), cancer stem cell (CSC), cancer-associated fibroblast (CAF), Neutrophil (Neu), tumour-associated macrophage (TAM), tumor-derived extracellular vesicles (TdEVs) (Made with BioRender.com, modified, accessed on 8 April 2022).

**Figure 2 cancers-14-02408-f002:**
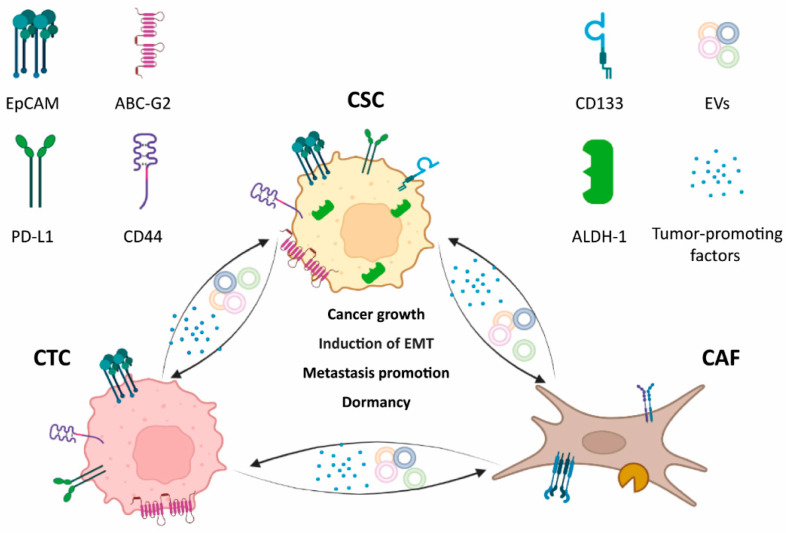
Key cellular players: cell interaction outcomes and well-known potential markers. Presented markers are only illustrative of generic cancer cells. Circulating tumor cell (CTC), cancer stem cell (CSC), cancer-associated fibroblast (CAF), tumor-derived extracellular vesicles (EVs). (Made with BioRender.com, modified, accessed on 8 April 2022).

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
