# Peer review of "Cancer Stem Cells (CSCs), Circulating Tumor Cells (CTCs) and Their Interplay with Cancer Associated Fibroblasts (CAFs): A New World of Targets and Treatments"

_cancers, 2022, doi:10.3390/cancers14102408_

Round 1

Reviewer 1 Report

In the manuscript presented by Aramini et al., authors aimed to review the current knowledge on cancer stem cells, circulating tumor cells and their interplay with cancer associated fibroblasts. The authors have put considerable effort into writing the manuscript and selecting the references in order to perform a very comprehensive review of current and interesting issues in the field.

Minor comments:

- thank you for addressing the notion of tumor cell plasticity as well as the reversible transformation of CSCs into cancerous cells but also of cancer cells into CSCs depending on the conditions and the micro-environment;

- in Figure 2, specify which type of tumor cells is represented? NSCLC? because all the tumor cells do not express all the markers represented (EpCAM, C44, CD133, etc.);

- EVs are mentioned several times and put on the same level as CSC, CTC, CAF (example line 449) but no paragraph is specifically dedicated to them. It could have been interesting to develop them more.

Author Response

Dear reviewer 1, thanks for all your comments which helped to improve our review article. Here attached it is the Rebuttal letter.

Thanks again.

Reviewer 2 Report

The authors report in this manuscript cancer stem cells (CSCs) and circulating tumor cells (CTCs). CSCs and CTCs are quite interesting theme. They referred to several reports concerning significance of CSCs and CTCs for diagnosis; functions of CSCs an CTCs in terms of progressive malignancy, i e. invasiveness, unique oligo cluster formation of CTCs facilitating metastasis.  They also report the significance of CSC in the activation of cancers occurred in the ECM rearrangements, and the opposite process to induce dormant cancers. CTCs are also important diagnostic markers for lung cancers potentiated distant metastases.

CSC/CTCs and CAFs participate crosstalk among cell populations to promote invasion metastasis and even promote dormancy. The clarification of CSC/CTCs would provide molecular targets for precision medicine.  

Comments;

The review is well written and reports quite interesting topics, however it should also report immunological topics, i e, immune escape of tumor cells, immune surveillance or immune checkpoint systems.

Full spelling of ALDH on line 284 ought to be on line 95, for the first time appearance.

Author Response

Dear reviewer 2,

many thanks for your comments, which have been helpful to improve our review article. Here attached it is the rebuttal Letter.

Thanks again.
